# MIMA: Iterative Model Averaging and Fine-Tuning for Multi-Task Learning

## Abstract

Fine-tuning large, pre-trained models on downstream tasks has become standard practice. But multi-task models that combine isolated task-specialised models remain challenging to construct. Task Arithmetic, a recent approach, merges multiple task-specific models into a single multi-task network simply by adding their "task vectors", without revisiting the original training data. In practice, model merging often results in substantial performance degradation. We show that independent fine-tuning of each model pushes these task vectors in orthogonal directions in parameter space. We hypothesise that actively aligning task vectors during fine-tuning will improve the performance of merged models. To test this hypothesis, we propose an iterative model averaging and fine-tuning framework called **MIMA**, which stands for **M**ulti-Task **I**terated **M**odel **A**veraging. We demonstrate that alternating phases of weight averaging and fine-tuning increase the pairwise cosine similarity between task vectors, encouraging knowledge sharing between tasks and preventing any one task vector from drifting too far from a unified model representation. When evaluated on a suite of eight vision benchmark tasks, MIMA retains competitive performance for each fine-tuned model on its single task, and significantly reduces the single-task accuracy gap between the fine-tuned model and the merged model to nearly zero, indicating the complete alignment between task vectors. Our work reveals new insights into the geometric relationship of the task vector in Task Arithmetic and presents a more effective framework for editing the behaviour of pre-trained models towards multi-task learning. [1]

## 1 Introduction

Foundation models, such as CLIP (Radford et al., 2021) and BERT (Devlin et al., 2019) have reshaped the default paradigm in machine learning (Bommasani et al., 2021). These large models, trained on large datasets, typically outperform or at least match smaller models whose architecture is specifically designed for a single task and trained from scratch, indicating that they capture generalisable representations for a large family of related tasks. To optimise performance on specific tasks, foundation, or other pre-trained models can be customised by fine-tuning, producing specialised models with excellent single-task performance (Yosinski et al., 2014; Kornblith et al., 2019). By design, fine-tuning on task-specific labelled data tends to overwrite earlier knowledge, leading to catastrophic forgetting and loss of generalisation (Kirkpatrick et al., 2017). Moreover, maintaining distinct fine-tuned models per task incurs substantial storage and deployment costs. Multi-task learning (MTL) addresses these issues through knowledge sharing across tasks. For traditional MTL, training a unified model on multiple tasks requires joint training and simultaneous access to all raw data, raising potential concerns about data privacy (Sanh et al., 2022).

As an alternative approach, model merging offers a cost-effective and scalable way to achieve high multi-task accuracy without data sharing. Given independently fine-tuned networks sharing a common initialisation, one can combine them directly in parameter space to obtain a unified multi-task model without accessing the original data (Wortsman et al., 2022a; Matena & Raffel, 2022). Different merging strategies employ linear interpolation between pre-trained and task-adapted weights (Ilharco et al., 2022), weighted averaging based on parameter importance (Matena & Raffel, 2022), or

---

[1]Large Language Models were used sparingly to polish the writing for certain paragraphs. All those generated outputs are further edited and revised by the authors before being used in the paper.

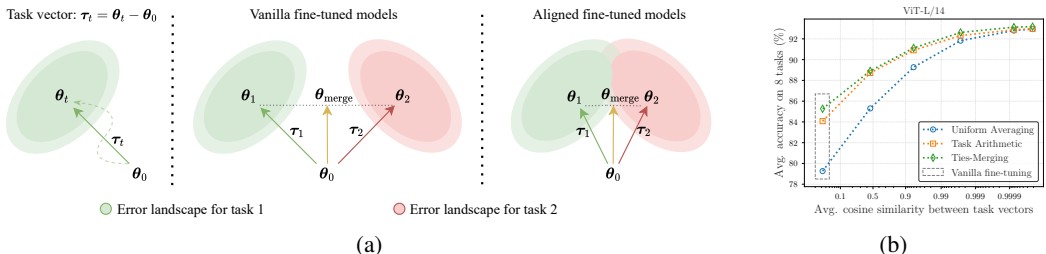

Figure 1: Illustration of the multi-task model merging through task vectors. **(a)** *Left:* The definition of task vector. The dashed line indicates the actual optimisation trajectory. *Middle:* Vanilla model fine-tuning, where models are fine-tuned independently, and their task vectors are merged post-fine-tuning. These vectors tend to be misaligned (orthogonal), causing the merged model to perform poorly across tasks as it lies far from any single-task optimum. *Right:* Alignment of task vectors increases cosine similarity between task vectors, leading the merged model closer to multiple task optima and improving overall multi-task performance. **(b)** Aligned task vectors yield higher average accuracy across eight tasks. Each marker represents a merged ViT-L/14 model derived from a set of task-specific models fine-tuned using MIMA with varying numbers of iterations under a fixed computational budget. As cosine similarity between final task vectors increases, both model merging methods show improved accuracy, and the accuracy advantage of Task Arithmetic over Uniform Average is reduced.

"Task Arithmetic" (Ilharco et al., 2023). In the latter, each fine-tuned model deviation defines a *task vector* (as illustrated in Fig. 1a (*left*)); summing these vectors with the pre-trained model produces the multi-task solution. Due to task interference, the independent fine-tuning of each task-specific model pushes these task vectors in orthogonal directions. Consequently, naive vector addition produces a merged model far from any individual task optimum (as depicted in Fig. 1a (*middle*)). This often degrades performance, evidenced by a high single-task accuracy drop from fine-tuned models to the merged model.

Instead of learning each task-specific model independently and only merging at the end, we introduce MIMA: **M**ulti-Task **I**terated **M**odel **A**veraging, which alternates between (i) weight averaging across all task-specific models and (ii) task-specific fine-tuning. By reinitialising each task-specific model iteratively from the weight-averaging one, we encourage all task updates to include a common component and facilitate knowledge transfer across tasks, thereby increasing the pairwise cosine similarity of their eventual task vectors. As a result, task vectors become more aligned, and their merged model lies closer to multiple task optima (as depicted in Fig. 1a (*right*)). We demonstrate that MIMA effectively *aligns* task vectors in each iteration, hence narrowing the single-task accuracy gap between the fine-tuned models and the merged model. Fig. 1b empirically shows that greater task vector alignment directly translates to higher multi-task accuracy. Merged models with higher cosine similarity between task vectors consistently outperform those obtained via vanilla fine-tuning across eight tasks. Moreover, the accuracy difference between the two merging methods diminishes as task vectors become better aligned.

Our main contributions are summarised as follows:

- We identify task-vector orthogonality as the bottleneck in model merging, and introduce MIMA to align task vectors during fine-tuning.

- We empirically validate MIMA on eight vision tasks and show MIMA with a larger number of iterations consistently outperforms vanilla fine-tuning methods in multi-task settings.

- We show that if task vectors are fully aligned, different merged methods yield similar performance, enabling Uniform Average to achieve results competitive with more complex methods like Task Arithmetic and Ties-Merging, thereby reducing the need for validation sets and hyperparameter selection.

## 2 RELATED WORK

**Multi-task learning.** MTL aims to improve the generalisation performance of a model by simultaneously learning multiple related tasks with a shared representation (Caruana, 1997). It has been applied to various problems in deep learning, from natural language processing (Collobert & Weston, 2008), speech recognition (Shinohara, 2016), to computer vision (Misra et al., 2016). MTL is typically conducted through *hard* or *soft* parameter sharing (Ruder, 2017). In *hard* parameter sharing, the parameters of the hidden layers are shared between tasks, while the parameters of the output layers are task-specific. It acts as a regularisation and reduces the risk of overfitting. Grad-Norm (Chen et al., 2018) was proposed to normalise gradient magnitudes when a single network is trained on multiple tasks. In contrast, *soft* parameter sharing maintains separate models for each task, but they are jointly connected, enabling cross-task knowledge transfer through learned connections. Misra et al. (2016) propose the "cross-stitch" units, which combine the activations from multiple networks trained on different tasks.

While conventional MTL methods assume simultaneous access to training data from all tasks, this assumption is often impractical in sensitive privacy settings where raw data cannot be shared across devices or institutions. To address this, we focus on a hard parameter sharing paradigm following the data privacy policy. Specifically, we fine-tune individual models on task-specific data independently and subsequently merge their parameters to construct a unified multi-task model. Crucially, our method only requires access and modification to the model parameters during fine-tuning, and does not necessitate sharing the original datasets.

**Federated learning.** Federated learning is a distributed computing paradigm where multiple clients train local models on their private data and collaborate to build one shared model. This is achieved by only sharing model updates from the clients, preserving data privacy. In federated learning, model merging occurs through multiple rounds of synchronisation among the clients. For example, Federated Averaging (FedAvg) (McMahan et al., 2017) maintains a single global model in the central server and optimises a single global objective. Each time, the central server sends a global model to a random fraction of clients to train for efficiency. Then, FedAvg iteratively performs model averaging across all clients and updates to the central server.

In contrast, MIMA maintains multiple task-specific objectives and maintains multiple task-specific models. All these task-specific models can be combined with any merging strategy (e.g., Uniform Averaging, Task Arithmetic, or Ties-Merging). More importantly, iterative averaging in MIMA is not used to enforce convergence to a single global model, but rather to align task vectors so that their merged multi-task model preserves single-task performance. Therefore, MIMA measures task-vector cosine similarity and the single-task accuracy gap of the merged model, while FedAvg measures the convergence of the training loss to a global model.

**Linear mode connectivity and model merging.** Models with the same initialisation or part of their optimisation path are situated within the same local basin. The accuracy does not decrease when linearly interpolating weights between them, dubbed as linear mode connectivity (LMC) (Izmailov et al., 2018; Frankle et al., 2020). LMC enables direct parameter manipulation within a shared basin and has been widely leveraged to merge models with the same architecture.

The first attempt for model merging with all models fine-tuned on the same task, aiming to improve its accuracy and generalisation. WiSE-FT (Wortsman et al., 2022b) computes a linear interpolation between the pre-trained parameters and the fine-tuned parameters. It shows large improvements in robustness under distribution shift, while preserving high accuracy on the target dataset. Fisher Merging (Matena & Raffel, 2022) uses the Fisher information (Fisher, 1922) to compute a weighted average of different models' parameters. Model Soup (Wortsman et al., 2022a) averages multiple fine-tuned models with different hyperparameter configurations and further improves the accuracy and robustness. Another approach attempts to merge models fine-tuned on different tasks to perform MTL, and this paper primarily focuses on the same objective. Ilharco et al. (2022) proposed to linearly interpolate the weights between fine-tuned models to build a multi-task model. RegMean (Jin et al., 2023) uses insights from linear models to minimise prediction differences between merged and individual models. However, this approach requires information from the dataset and needs to compute the inner product matrix for the training dataset.

Task Arithmetic (Ilharco et al., 2023) introduces the concept of "task vector". It builds the multi-task model by adding and scaling all task vectors to the pre-trained weights. One disadvantage of this method is its reliance on a scaling term, which requires optimisation. We show that when task vectors are aligned, simple averaging matches the performance of task-vector learning, hence eliminating the need for a scaling term. The main limitation of task arithmetic methods, however, arises from interference between task vectors. A variety of works have attempted to reduce such interference, but despite some improvement, the accuracy gap between the single-task fine-tuned models and the multi-task merged model remains large. For example, Ortiz-Jiménez et al. (2023) proposed weight disentanglement, which allows a model to perform task arithmetic by independently manipulating these distinct task vector directions. They showed that fine-tuning models in their tangent space amplifies this weight disentanglement property, leading to better performance of the merged models. TIES-Merging (Yadav et al., 2023) shows that removing redundant parameters and reducing the sign conflict between task vectors improves the performance of the merged models. Similarly to these approaches, MIMA exploits the error landscape. Our approach is iterative: MIMA alternately improves single-task performance (through single-task fine-tuning, increasing the interference) and optimises the merged model (through repeated model averaging). We show that iterating this combination of steps aligns task vectors to effectively suppress interference between them. Hence, MIMA provides a combination of high single-task and multi-task accuracy.

## 3 PROBLEM STATEMENT

**Notation and vanilla fine-tuning.** Let $\boldsymbol{\theta}_0 \in \mathbb{R}^d$ represent the weights of the pre-trained model, where $d$ is the number of parameters. We consider a set of $T$ downstream tasks, indexed by $t \in \{1, \dots, T\}$, each task $t$ with its own labelled dataset $\mathcal{D}_t$. In vanilla fine-tuning, a separate model is trained from $\boldsymbol{\theta}_0$ for task $t$, producing task-specific weights $\boldsymbol{\theta}_t$:

$$\boldsymbol{\theta}_t = \text{FineTune}(\boldsymbol{\theta}_0, \mathcal{D}_t, S),$$

where $S$ is the total number of gradient-descent steps allocated per task.

**Task Arithmetic.** Following (Ilharco et al., 2023), the task vector $\boldsymbol{\tau}_t$ for task $t$ is defined as the vector difference between the fine-tuned weights and pre-trained weights. Mathematically,

$$\boldsymbol{\tau}_t = \boldsymbol{\theta}_t - \boldsymbol{\theta}_0. \tag{1}$$

This task vector $\boldsymbol{\tau}_t$ represents the weight update through fine-tuning in parameter space. Therefore, a merged multi-task model can be obtained by simply adding the sum of each task vector to the pre-trained model:

$$\boldsymbol{\theta}_{\text{merge}} = \boldsymbol{\theta}_0 + \lambda \sum \boldsymbol{\tau}_t = \boldsymbol{\theta}_0 + \lambda \sum_{t=1}^{T} (\boldsymbol{\theta}_t - \boldsymbol{\theta}_0), \tag{2}$$

where $\lambda$ is the scaling factor determined using held-out validation sets from $\{\mathcal{D}_t\}_{t=1}^{T}$. When $\lambda = 1/T$, the resulting weights $\boldsymbol{\theta}_{\text{merge}}$ are the same as the average of the fine-tuned weights across all tasks, i.e., $\boldsymbol{\theta}_{\text{merge}} = \sum_{t=1}^{T} \boldsymbol{\theta}_t / T$.

**Interference via orthogonal task vectors.** Since Task Arithmetic does a linear combination between task vectors on the basis of a pre-trained model, the efficacy of the merging model in equation 2 depends on the geometric relationship between the task vectors and error landscapes. If the merged model is in the basin for each task, it can achieve high multi-task accuracy. In practice, independently fine-tuning models on distinct tasks often finds that task vectors lie in near-orthogonal directions (see Appendix C.1):

$$\frac{\boldsymbol{\tau}_i \cdot \boldsymbol{\tau}_j}{\|\boldsymbol{\tau}_i\| \|\boldsymbol{\tau}_j\|} \approx 0, \ \forall i \neq j \tag{3}$$

Such orthogonality leads to task interference when summing $\{\boldsymbol{\tau}_t\}$, as the merged model $\boldsymbol{\theta}_{\text{merge}}$ may lie in a region of the error landscape far from any individual task optimum, causing large accuracy drops on individual tasks.

We aim to design a fine-tuning procedure that, under a fixed total fine-tuning budget $S$ per task, generates a set of aligned task vectors $\{\boldsymbol{\tau}_t\}_{t=1}^{T}$. By implicitly encouraging high pairwise cosine similarity between the task vectors during training, we could build an effective multi-task model through a simple merge method (e.g. Uniform Averaging) without using any complex post-hoc modifications (TIES-Merging) or hyperparameter tuning (Task Arithmetic).

# 4 MIMA

We introduce MIMA and describe its detailed implementation in this section. We split the entire fine-tuning process into multiple iterations. Each iteration consists of a uniform-averaging phase followed by a fine-tuning phase, as illustrated in Fig 2. Instead of learning in isolation, our method frequently performs synchronisation across all task models and iteratively changes the starting point of fine-tuning, ensuring knowledge sharing among task-specific models.

**Uniform-averaging phase.** At the beginning of each iteration $i \in \{2, \ldots, N\}$, we perform a synchronisation step across all $T$ task-specific models before the fine-tuning phase. This phase establishes a shared representation that integrates knowledge accumulated from all tasks encountered thus far. Specifically, we compute the average of the task-specific weights from the previous iteration, which serves as a common starting point for all tasks in the fine-tuning phase:

$$\bar{\boldsymbol{\theta}}^{(i-1)} = \frac{1}{T} \sum_{t=1}^{T} \boldsymbol{\theta}_t^{(i-1)}, i \geq 2 \qquad (4)$$

For the first iteration ($i = 1$), we initialise $\bar{\boldsymbol{\theta}}^{(0)}$ with the pretrained model: $\bar{\boldsymbol{\theta}}^{(0)} = \boldsymbol{\theta}_0$.

**Fine-tuning phase.** Following the uniform-averaging phase, each task-specific model $\boldsymbol{\theta}_t$ is fine-tuned on its labelled dataset, starting from the shared representation $\bar{\boldsymbol{\theta}}^{(i-1)}$, for $K$ optimisation steps ($K \leq S$):

$$\boldsymbol{\theta}_t^{(i)} = \text{FineTune}(\bar{\boldsymbol{\theta}}^{(i-1)}, \mathcal{D}_t, K).$$

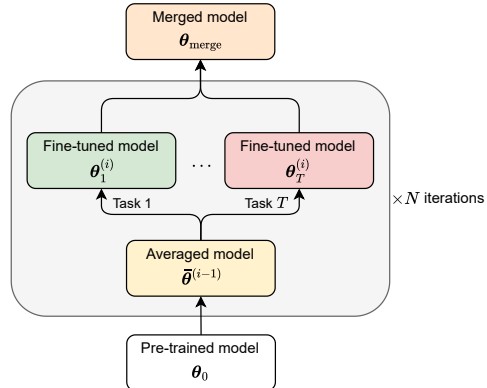

Figure 2: MIMA framework: A common pre-trained model undergoes $N$ iterations of uniform-averaging over the weights and fine-tuning phases for each task. Finally, task-specific models are merged. The $N = 1$ case is equivalent to vanilla model merging.

We define the incremental update $\boldsymbol{\delta}_t^{(i)}$ for task $t$ at iteration $i$ as the parameter change before and after the fine-tuning phase:

$$\boldsymbol{\delta}_t^{(i)} = \boldsymbol{\theta}_t^{(i)} - \bar{\boldsymbol{\theta}}^{(i-1)}. \qquad (5)$$

While the task vector $\boldsymbol{\tau}_t$ represents the total cumulative parameter change relative to the initial model $\boldsymbol{\theta}_0$, the incremental update $\boldsymbol{\delta}_t$ is the local parameter change that occurs from the current fine-tuning phase. Importantly, these incremental updates $\boldsymbol{\delta}_t$ tend to remain orthogonal, as they result from independent training on distinct tasks. Therefore, we update equation 3 to the following:

$$\frac{\boldsymbol{\delta}_i \cdot \boldsymbol{\delta}_j}{\|\boldsymbol{\delta}_i\|\|\boldsymbol{\delta}_j\|} \approx 0, \forall i \neq j \qquad (6)$$

In contrast, task vectors may become increasingly aligned over iterations due to accumulated shared components as discussed later. Fig. 3 illustrates the distinction between the local incremental update $\boldsymbol{\delta}_t$ and the global task vector $\boldsymbol{\tau}_t$, and we empirically validate such behaviour of incremental updates and task vectors over iterations in Appendix C.2.

**Iteration.** The above weight-averaging and fine-tuning phases are repeated for $N$ iterations. To ensure a fair comparison, the total fine-tuning steps $S$ is fixed:

$$S = K \times N.$$

Increasing $N$ introduces more frequent synchronisation at the cost of fewer weight update steps per phase, offering a tunable trade-off between exploration (task-specific adaptation) and alignment (cross-task knowledge sharing).

The averaged model at iteration $i$ is the initial pre-trained model plus the average of all incremental updates up to that point. Combining equation 4 and equation 5, the evolution of the averaged model can be expressed as:

$$\bar{\boldsymbol{\theta}}^{(i-1)} = \frac{1}{T} \sum_{t=1}^{T} \boldsymbol{\theta}_t^{(i-1)} = \bar{\boldsymbol{\theta}}^{(i-2)} + \frac{1}{T} \sum_{t=1}^{T} \boldsymbol{\delta}_t^{(i-1)}$$

$$= \boldsymbol{\theta}_0 + \frac{1}{T} \sum_{l=1}^{i-1} \sum_{t=1}^{T} \boldsymbol{\delta}_t^{(l)}, i \geq 2 \tag{7}$$

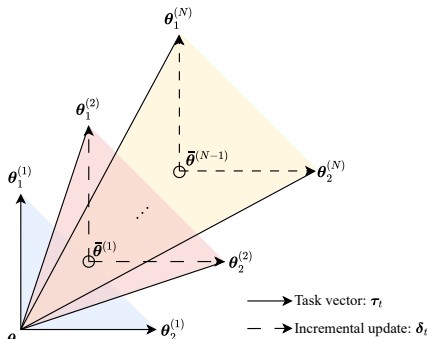

Figure 3: A vector-based illustration of the update process for two tasks. The solid line represents the task vector $\boldsymbol{\tau}_t$ while the dashed line represents the incremental update $\boldsymbol{\delta}_t$. At the first iteration, $\boldsymbol{\tau}_t^{(1)} = \boldsymbol{\delta}_t^{(1)}$. Over iterations, shared components $\bar{\boldsymbol{\theta}}$ accumulate, aligning $\boldsymbol{\tau}_t$ despite $\boldsymbol{\delta}_t$ remaining near-orthogonal.

Accordingly, the task vector $\boldsymbol{\tau}_t^{(i)}$ after iteration $i$ can be decomposed as:

$$\boldsymbol{\tau}_t^{(i)} = \begin{cases} \boldsymbol{\delta}_t^{(1)}, & \text{if } i = 1; \\ \frac{1}{T} \sum_{l=1}^{i-1} \sum_{t=1}^{T} \boldsymbol{\delta}_t^{(l)} + \boldsymbol{\delta}_t^{(i)}, & \text{otherwise} \end{cases} \tag{8}$$

This highlights that each task vector contains both a task-specific update term $\boldsymbol{\delta}_t^{(i)}$ and an accumulating shared term $\frac{1}{T} \sum_{l=1}^{i} \sum_{t=1}^{T} \boldsymbol{\delta}_t^{(l)}$. As iteration progresses, the shared term becomes dominant, leading to implicit alignment across task vectors.

**Final merge.** After $N$ iterations, we obtain a set of task-specific weights $\{\boldsymbol{\theta}_t^{(N)}\}_{t=1}^{T}$. These can then be merged into unified weights $\theta_{\text{merge}}$ using any standard merging method.

# 5 EMPIRICAL INVESTIGATION: MIMA

In this section, we present the experimental settings (Sec. 5.1) and results (Sec. 5.2). Additional details and results are included in the Appendix due to page limitations.

## 5.1 EXPERIMENTAL SETTINGS

**Fine-tuning.** All the fine-tuning experiments follow the same training procedure specified in Ilharco et al. (2022; 2023); Ortiz-Jiménez et al. (2023). We start with the same pre-trained CLIP ViT model (ViT-B/32, ViT-B/16, ViT-L/14) downloaded from the `open_clip` repository (Ilharco et al., 2021) as our pre-trained model $\boldsymbol{\theta}_0$ and fine-tune it through end-to-end supervised learning for a total of $S = 2000$ iterations on each task. We use the AdamW optimiser with a batch size of 128, a learning rate of 1e-5, a weight decay (Loshchilov & Hutter, 2019) of 0.1, and a cosine annealing learning rate schedule with 200 warm-up steps. When fine-tuning, we froze the final classification layer output and only optimise the image encoder layers. We vary the number of MIMA iterations ($N$) across $\{1, 2, 4, 8, 16, 20\}$. Note that the case $N = 1$ corresponds to the standard, independent fine-tuning baseline.

**Datasets.** We use eight published image classification datasets for our eight tasks, including Cars (Krause et al., 2013), DTD (Cimpoi et al., 2014), EuroSAT (Helber et al., 2019), GTSRB (Stallkamp et al., 2011), MNIST (LeCun, 1998), RESISC45 (Cheng et al., 2017), SUN397 (Xiao et al., 2016),

and SVHN (Netzer et al., 2011). We apply default settings for training and testing, using each validation set for hyperparameter selection including the Task Arithmetic scaling factor $\lambda$.

**Merging method.** We evaluate three primary merging methods after the final MIMA iteration: (1) **Uniform Averaging** calculates the element-wise mean over the weights of fine-tuned models: $\frac{1}{T}\sum_{t=1}^{T}\boldsymbol{\theta}_t^{(N)}$. (2) **Task Arithmetic** (Ilharco et al., 2023) scales and sums the final task vectors and the initial pre-trained model: $\boldsymbol{\theta}_0 + \lambda\sum_{t=1}^{T}(\boldsymbol{\theta}_t^{(N)} - \boldsymbol{\theta}_0)$. The optimal scaling factor $\lambda$ is found via a grid search over $[0, 0.05, 0.1, \ldots, 1.0]$ on held-out validation sets. (3) **Ties-Merging** (Yadav et al., 2023) follows three steps of emphTrim, *Elect Sign*, and *Disjoint Merge* on task vectors. The optimal scaling factor $\lambda$ is found via a grid search over $[0.8, 0.85, 0.9, \ldots, 1.8]$ on held-out validation sets.

We also evaluate a practical setting where no validation set is available. Uniform Averaging is a hyperparameter-free method without the need for a validation set. Following the recommendations from previous work, we set $\lambda = 1$ for Ties-Merging and $\lambda = 0.4$ for Task Arithmetic.

## 5.2 EXPERIMENTAL RESULTS

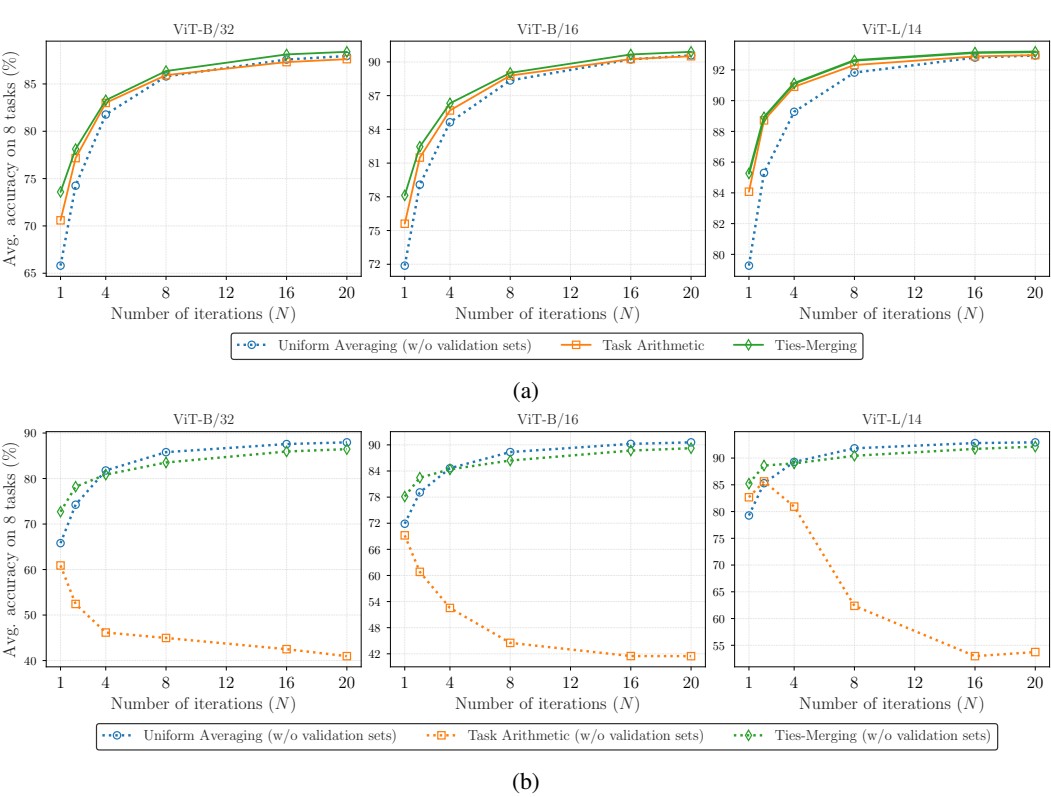

(a)

(b)

Figure 4: Comparing the average absolute accuracy on eight tasks among Uniform Averaging, Task Arithmetic and Ties-Merging over varying numbers of iterations $N$. Dotted lines indicate results *without* validation sets, while solid lines indicate results *with* validation sets. **(a)**: With validation sets, Task Arithmetic and Ties-Merging benefit from additional numbers of iterations. **(b)**: Without validation sets, Task Arithmetic is unstable and its performance drops substantially as $N$ increases.

**MIMA performance.** Fig. 4 illustrates the trend of average absolute accuracy on the eight vision tasks as a function of the number of iterations $N$, comparing Uniform Averaging, Task Arithmetic and Ties-Merging for the three ViT architectures. The performance of Uniform Averaging and Task Arithmetic improves monotonically with a larger number of iterations $N$ across all architectures. For Fig. 4a, after selecting the optimal scaling term $\lambda$ for task vectors using validation sets, both Task Arithmetic and Ties-Merging achieve higher average accuracy than Uniform Averaging. However, their performance advantage over Uniform Averaging diminishes as $N$ increases. This indicates that

with sufficient alignment among task vectors, a simple, hyperparameter-free averaging of weights is nearly as effective as the more complex approaches. Fig. 4b evaluates merging in the absence of the validation set. In this case, neither Task Arithmetic nor Ties-Merging can consistently outperform Uniform Averaging when more frequent MIMA averaging is used. In fact, the alignment between task vectors can hurt Task Arithmetic with a fixed scaling term $\lambda$, leading to degraded performance as the number of iterations grows.

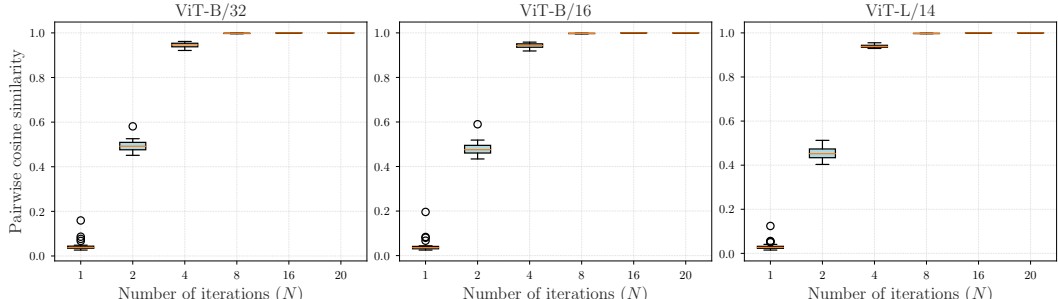

Figure 5: Box plot for pairwise cosine similarities between final task vectors as a function of the number of iterations $N$.

**Pairwise cosine similarity.** To directly validate our central hypothesis, we compute the pairwise cosine similarity between the final task vectors $\tau_t^{(N)}$. Fig. 5 shows the distribution of these similarities as $N$ increases. For the baseline case of vanilla fine-tuning ($N = 1$), the pairwise cosine similarities are very low, typically centred around 0.0 to 0.1, confirming the near-orthogonality of independently learned task vectors. As $N$ increases, the median and interquartile range of cosine similarities shift steadily towards 1. This provides direct evidence that MIMA successfully aligns task vectors, forcing them to point in a common direction in parameter space.

**Single-task accuracy gap.** A key challenge in multi-task learning is preserving the high performance of task-specific models. We define the single-task accuracy gap as the difference between the accuracy of an individual fine-tuned model and the accuracy of the merged model on that same task: $\mathrm{Acc}(\boldsymbol{\theta}_t^{(N)}, \mathcal{D}_t) - \mathrm{Acc}(\boldsymbol{\theta}_{\mathrm{merge}}, \mathcal{D}_t)$. Fig. 6 shows that this gap shrinks to nearly zero over 16 iterations, demonstrating that the single multi-task model performs almost as well as a collection of task-specific models.

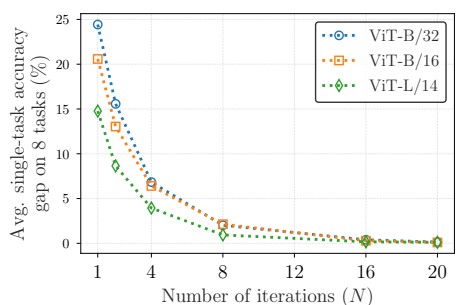

Figure 6: Average single-task accuracy gap between individual fine-tuned models and the merged model through Uniform Averaging with varying numbers of iterations $N$.

**Error landscape visualisation.** To build a geometric intuition for the effectiveness of MIMA, we visualise the error landscape for fine-tuning ViT-B/16 models with varying numbers of iterations. Fig. 7 plots a 2D slice of the error landscape for two tasks (Cars and RESISC45). The plane is defined by the pre-trained model and two task-specific models fine-tuned on each task.

For vanilla fine-tuning baseline ($N = 1$), the low-error "valleys" for the two tasks are far apart, and the average model may lie in a region of high error for both. As we increase $N$ to 4 and then 8, the two found task-specific valleys rotate towards each other. At $N = 8$, the two valleys are nearly parallel, creating a basin of shared low error. The average model now could reside within this shared basin, achieving low error for both tasks simultaneously. This visualisation confirms that MIMA could find a more suitable landscape to resolve task conflict with a shared solution space.

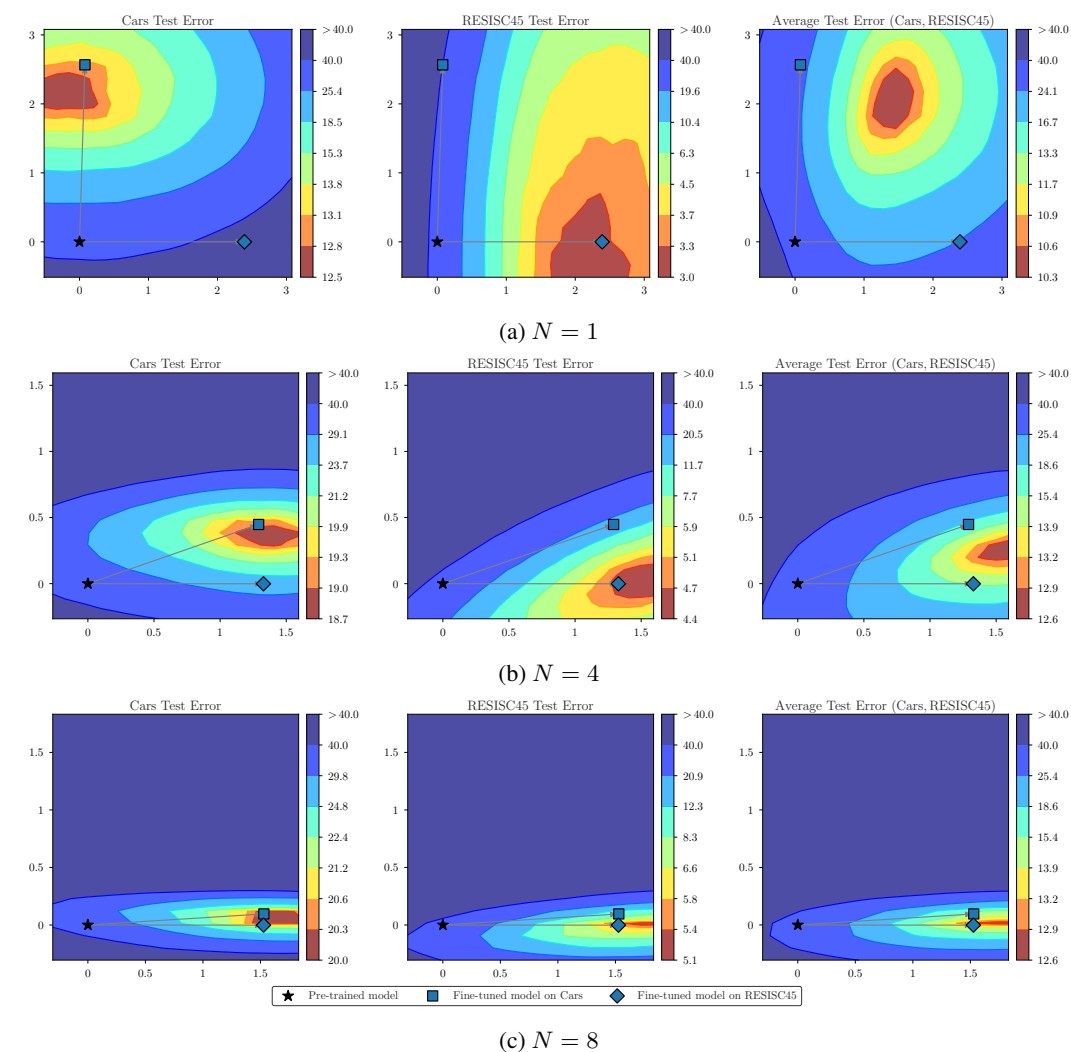

(a) $N = 1$

(b) $N = 4$

(c) $N = 8$

Figure 7: Increasing the number of iterations in MIMA decreases the angle between final task vectors (illustrated by the grey arrows). This figure shows a two-dimensional slice of the error landscapes for ViT-B/16 models on Cars (*left*), RESISC45 (*middle*), and their average (*right*). As in Izmailov et al. (2018), we obtain the orthonormal basis $\boldsymbol{u}, \boldsymbol{v}$ for the plane spanned by the two task vectors. A point $P$ with coordinates $(x, y)$ in the plane would then be given by $P = \boldsymbol{\theta}_0 + x \cdot \boldsymbol{u} + y \cdot \boldsymbol{v}$.

## 6 CONCLUSION

In this paper, we addressed a challenge in multi-task model merging: the task interference caused by combining independently trained models with near-orthogonal task vectors. We proposed MIMA, a framework that interleaves vanilla fine-tuning with iterative weight averaging to enforce alignment between task-specific models during training.

Our empirical results on vision tasks confirm this, demonstrating that our algorithm dramatically improves the performance of merged models. By progressively aligning task vectors, our algorithm enables a single multi-task model to retain nearly all of the performance of its specialised, fine-tuned models, effectively reducing the single-task accuracy gap. The simplicity and effectiveness of our proposed algorithm make it a practical and powerful technique for building efficient and capable multi-task systems from large pre-trained foundation models.

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

## A  EXPERIMENTAL SETTINGS

### A.1  VISION DATASET DETAILS

- **Cars** (Krause et al., 2013) contains car images across 196 classes. The train/validation/test splits have $7,333/814/8,041$ images, respectively.
- **DTD** (Cimpoi et al., 2014) contains texture images "in the wild" in 47 classes. The train/validation/test splits have $3,384/376/1,880$ images, respectively.
- **EuroSAT** (Helber et al., 2019) contains Sentinel-2 satellite images in 10 classes. The train/validation/test splits have $21,600/2,700/2,700$ images, respectively.
- **GTSRB** (Stallkamp et al., 2011) contains images of traffic sign in 43 classes. The train/validation/test splits have $23,976/2,664/12,630$ images, respectively.
- **MNIST** (LeCun, 1998) contains images of handwritten digit in 10 classes. The train/validation/test splits have $55,000/5,000/10,000$ images, respectively.
- **RESISC45** (Cheng et al., 2017) contains images of remote sensing scene in 45 classes. The train/validation/test splits have $17,010/1,890/6,300$ images, respectively.
- **SUN397** (Xiao et al., 2016) contains scene images in 397 classes. The train/validation/test splits have $17,865/1,985/19,850$ images, respectively.
- **SVHN** (Netzer et al., 2011) contains images of printed digits cropped from house number plates in 10 classes. The train/validation/test splits have $68,257/5,000/26,032$ images, respectively.

For any dataset without a publicly available test set, we use its validation set for testing and re-split the original training set into train/validation subsets. For all datasets, we report classification accuracy as our evaluation metric.

### A.2  COMPUTING INFRASTRUCTURE

All fine-tuning and evaluation of each CLIP ViT model are conducted on a single NVIDIA A100 GPU using PyTorch, running on an internal compute cluster.

## B  ADDITIONAL PERFORMANCE RESULTS

In this section, we present additional experiments to expand the findings discussed in the main text.

### B.1  MAIN RESULTS

Table 1 provides a detailed comparison of merging methods across three CLIP ViT architectures, showing both average absolute and relative accuracies as the number of MIMA iterations ($N$) increases. Specifically, we define the relative accuracy for the merged model as the single-task accuracies achieved by the merged model divided by the single-task accuracies achieved by the corresponding individual fine-tuned model (averaged across all tasks). Mathematically,

$$\text{Relative accuracy} = \frac{1}{T} \sum_{t=1}^{T} \frac{\text{Acc}(\boldsymbol{\theta}_{\text{merge}}, \mathcal{D}_t)}{\text{Acc}(\boldsymbol{\theta}_t, \mathcal{D}_t)}$$

By this definition, the relative accuracy of the individual fine-tuned models on their respective tasks is 100%.

The results for the baseline case ($N = 1$) show a large performance gap between the individually fine-tuned models and the merged models. For ViT-L/14, Task Arithmetic and Ties-Merging achieve

Table 1: Average absolute (%) and relative accuracies (%) of different CLIP ViTs merged by Uniform Averaging, Task Arithmetic and Ties-Merging on 8 tasks. We report results for varying numbers of MIMA iterations ($N$), and $N = 1$ corresponds to the baseline vanilla fine-tuning approach.

| Method | Number of Iterations | ViT-B/32 | | ViT-B/16 | | ViT-L/14 | |
| --- | --- | --- | --- | --- | --- | --- | --- |
| | | Abs. (↑) | Rel. (↑) | Abs. (↑) | Rel. (↑) | Abs. (↑) | Rel. (↑) |
| Pre-trained | - | 48.1 | – | 55.3 | – | 65.2 | – |
| Fine-tuned | | 90.2 | 100 | 92.5 | 100 | 94.0 | 100 |
| Uniform Averaging | $N = 1$ | 65.8 | 73.3 | 71.9 | 77.7 | 79.3 | 84.2 |
| Task Arithmetic | | 70.6 | 79.4 | 75.6 | 82.4 | 84.1 | 89.9 |
| Ties-Merging | | 73.6 | 82.7 | 78.1 | 85.2 | 85.3 | 91.2 |
| Fine-tuned | | 89.8 | 100 | 92.1 | 100 | 93.9 | 100 |
| Uniform Averaging | $N = 2$ | 74.2 | 82.7 | 79.1 | 85.6 | 85.3 | 90.6 |
| Task Arithmetic | | 77.2 | 87.2 | 81.5 | 89.2 | 88.7 | 94.9 |
| Ties-Merging | | 78.1 | 88.3 | 82.5 | 90.3 | 88.9 | 95.1 |
| Fine-tuned | | 88.6 | 100 | 91.0 | 100 | 93.2 | 100 |
| Uniform Averaging | $N = 4$ | 81.8 | 92.1 | 84.6 | 92.7 | 89.3 | 95.6 |
| Task Arithmetic | | 83.0 | 95.4 | 85.7 | 95.1 | 90.9 | 98.2 |
| Ties-Merging | | 83.3 | 95.8 | 86.3 | 95.8 | 91.1 | 98.4 |
| Fine-tuned | | 87.8 | 100 | 90.5 | 100 | 92.8 | 100 |
| Uniform Averaging | $N = 8$ | 85.8 | 97.7 | 88.4 | 97.6 | 91.8 | 98.9 |
| Task Arithmetic | | 85.9 | 99.9 | 88.8 | 99.2 | 92.3 | 100.2 |
| Ties-Merging | | 86.4 | 100.4 | 89.0 | 99.5 | 92.6 | 100.5 |
| Fine-tuned | | 88.0 | 100 | 90.5 | 100 | 93.0 | 100 |
| Uniform Averaging | $N = 16$ | 87.6 | 99.5 | 90.2 | 99.7 | 92.8 | 99.8 |
| Task Arithmetic | | 87.3 | 101.1 | 90.3 | 100.8 | 92.9 | 100.6 |
| Ties-Merging | | 88.1 | 102.1 | 90.7 | 101.3 | 93.1 | 100.8 |
| Fine-tuned | | 88.1 | 100 | 90.7 | 100 | 93.1 | 100 |
| Uniform Averaging | $N = 20$ | 88.0 | 99.8 | 90.6 | 99.9 | 92.9 | 99.9 |
| Task Arithmetic | | 87.6 | 101.3 | 90.5 | 100.9 | 93.0 | 100.5 |
| Ties-Merging | | 88.4 | 102.2 | 90.9 | 101.3 | 93.2 | 100.8 |

Table 2: The optimal scaling term $\lambda$ used in Task Arithmetic, selected via a held-out validation.

| | $N = 1$ | $N = 2$ | $N = 4$ | $N = 8$ | $N = 16$ | $N = 20$ |
| --- | --- | --- | --- | --- | --- | --- |
| ViT-B/32 | 0.25 | 0.2 | 0.15 | 0.15 | 0.15 | 0.15 |
| ViT-B/16 | 0.25 | 0.2 | 0.15 | 0.15 | 0.15 | 0.15 |
| ViT-L/14 | 0.3 | 0.25 | 0.2 | 0.15 | 0.15 | 0.15 |

Table 3: The optimal scaling term $\lambda$ used in Ties-Merging, selected via a held-out validation.

| | $N = 1$ | $N = 2$ | $N = 4$ | $N = 8$ | $N = 16$ | $N = 20$ |
| --- | --- | --- | --- | --- | --- | --- |
| ViT-B/32 | 0.85 | 1.05 | 1.45 | 1.5 | 0.45 | 1.5 |
| ViT-B/16 | 1 | 1 | 1.5 | 1.7 | 1.6 | 1.5 |
| ViT-L/14 | 1.05 | 1.3 | 1.8 | 1.7 | 1.7 | 1.65 |

84.1% and 85.3% average absolute accuracy, respectively, while compared to the absolute average accuracy of each fine-tuned model on its own task is 94.0%. This gap is even larger for Uniform Averaging, which achieves 79.3% average absolute accuracy. This is expected, as Task Arithmetic and Ties-merging use a tuned scaling factor $\lambda$ to partially mitigate misalignment (see optimal values in Tables 2 and 3), whereas Uniform Averaging does not.

With MIMA, the performance of the merged models improves dramatically. As $N$ increases, both Uniform Averaging and Task Arithmetic see substantial accuracy gains. For ViT-L/14, the average absolute accuracy of the model merged with Task Arithmetic also sees a similar rise from 84.1% ($N = 1$) to 93.0% ($N = 20$). Similarly, Uniform Averaging increases from 79.3% to 92.9%. This demonstrates that our method effectively mitigates task interference, allowing even a naive averaging approach to produce a high-performing multi-task model. Notably, with sufficient iterations ($N \geq 8$), the merged models achieve relative accuracies near or even exceeding 100%, indicating that the final multi-task model is on par with, and sometimes even better than, the task-specific models.

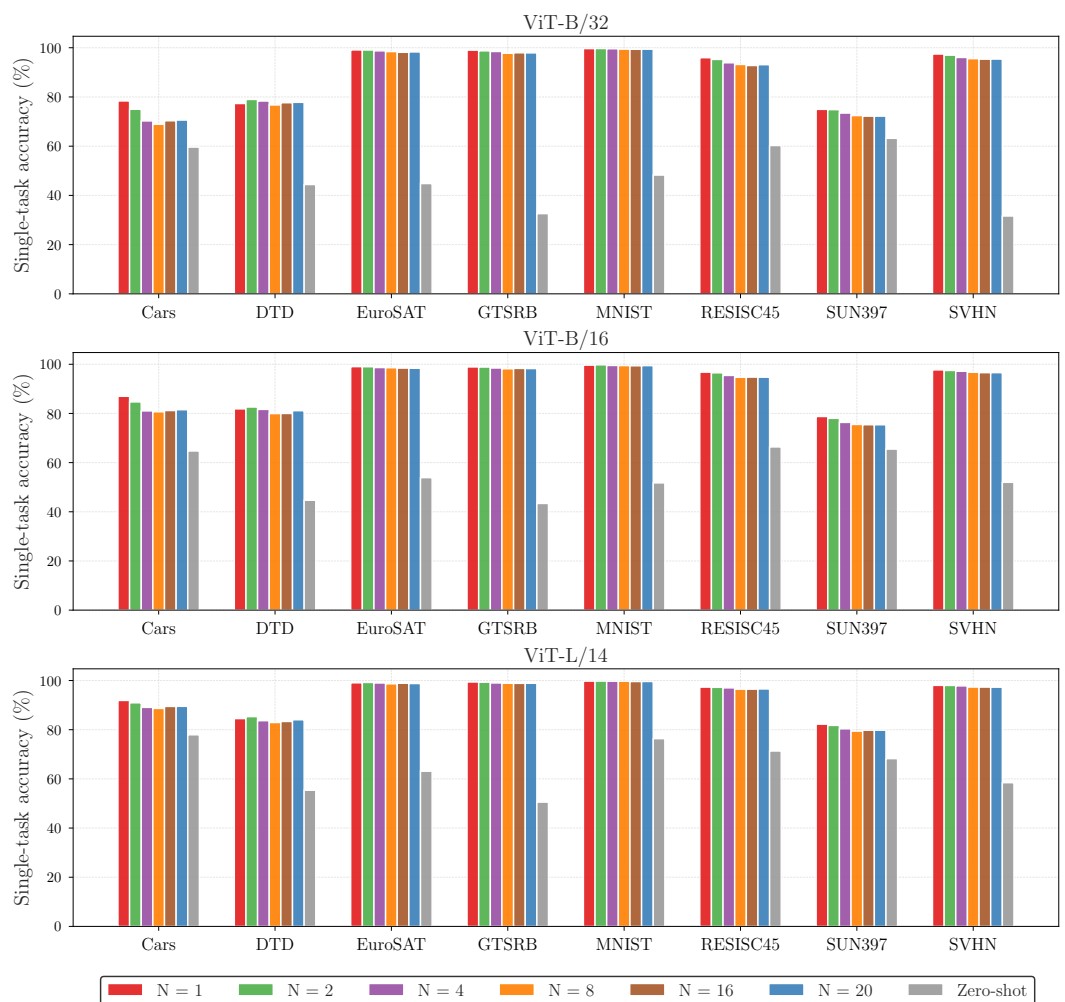

Figure 8: Single-task accuracy for zero-shot model and each task-specific model with different numbers of iterations.

## B.2 FINE-TUNING ACCURACIES

A key benefit of our method is that it improves the multi-task accuracy without sacrificing the single-task specialisation. Figure 8 shows the single-task accuracies achieved by task-specific models: before fine-tuning (referred to as *zero-shot*) and after fine-tuning with different numbers of iterations. These results indicate that an increase in the number of iterations will not significantly change the single-task accuracy for any given task.

### B.3 MULTI-TASK ACCURACIES

Figure 9 reports the absolute accuracies of the merged models using Uniform Averaging. The performance of the merged model improves uniformly across all eight tasks with a larger number of MIMA iterations.

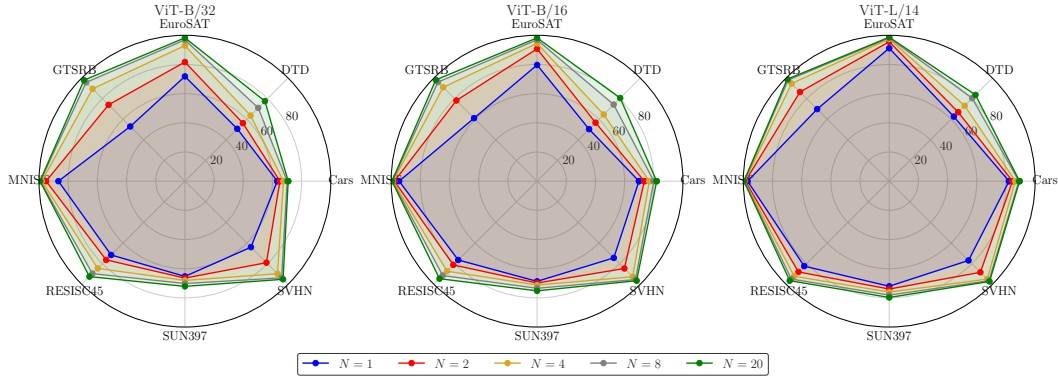

Figure 9: Comparison of the merged multi-task model using Uniform Average on eight tasks. The multi-task accuracy is uniformly improved as the number of iterations increases.

## C ADDITIONAL COSINE SIMILARITY RESULTS

In this section, we evaluate the cosine similarity between incremental updates $\delta_t$ or task vectors $\tau_t$ for the fine-tuned models during the iteration.

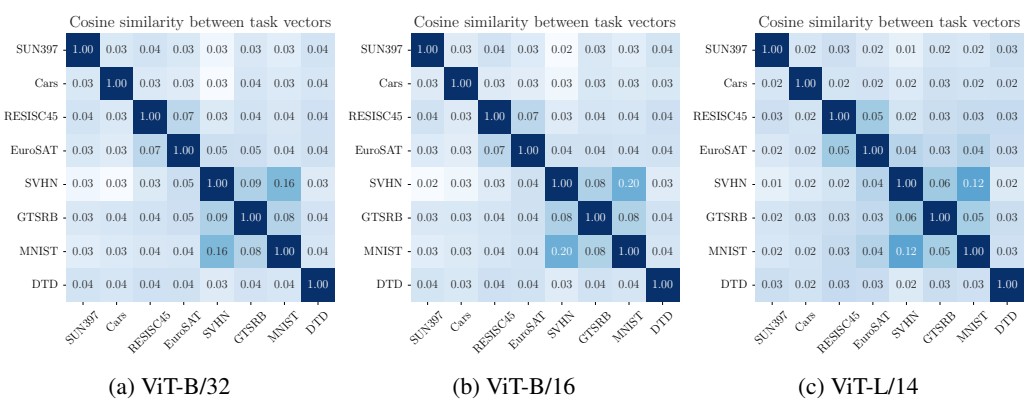

Figure 10: The cosine similarity matrix for the task vectors obtained from vanilla fine-tuning CLIP models ($N = 1$). Normally, most task vectors are nearly orthogonal (similarity $\approx 0$), indicating misalignment. Two semantically similar tasks will slightly increase the cosine similarity on the task vector, such as MNIST and SVHN.

### C.1 VANILLA FINE-TUNING ($N = 1$)

According to the definition, the incremental updates and task vectors are the same $\delta_t^{(1)} = \tau_t^{(1)}$ for the baseline case $N = 1$. Figure 10 shows the pairwise cosine similarity matrix, where most pairs have a similarity close to zero. Two tasks with semantic similarity (MNIST and SVHN) produce task vectors with a larger cosine similarity. The observations have also been observed by (Ilharco et al., 2023).

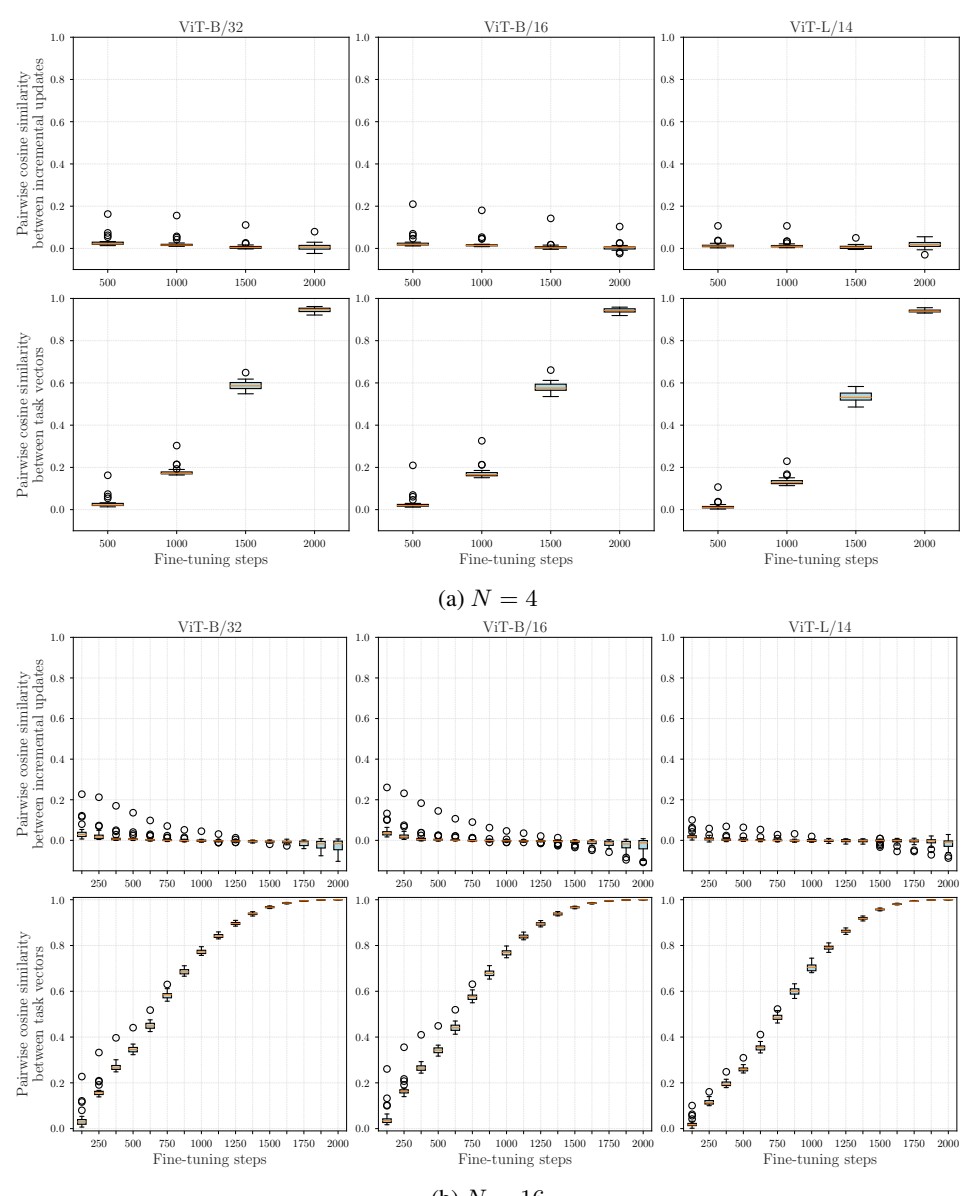

(a) $N = 4$

(b) $N = 16$

Figure 11: Evolution of the cosine similarity between any task vector $\boldsymbol{\tau}_t$ or incremental update $\boldsymbol{\delta}_t$ during fine-tuning for (**a**) $N = 4$ and (**b**) $N = 16$ MIMA iterations. Each box plot shows the distribution of pairwise cosine similarity at a given fine-tuning step, which is at the end of the fine-tuning phase. The cosine similarity of the incremental update $\boldsymbol{\delta}_t$ remains near 0 while the task vector $\boldsymbol{\tau}_t$ steadily increases towards 1.

## C.2 MIMA ($N > 1$)

With MIMA, the behaviours of task vectors $\boldsymbol{\tau}_t$ and incremental updates $\boldsymbol{\delta}_t$ diverge during fine-tuning. Figure 11 shows this phenomenon for $N = 4$ and $N = 16$. At the end of each fine-tuning phase, the incremental update $\boldsymbol{\delta}_t$ between any different tasks remains nearly orthogonal to each other. However, the cosine similarity between any task vectors $\boldsymbol{\tau}_t$ gradually increases with each iteration. As the total number of fine-tuning steps approaches the budget (20000 fine-tuning steps), the cosine similarity between task vectors approaches 1. This observation validates our hypothesis described in the main content, where MIMA forces task vectors into a shared direction in the parameter space.

# D ADDITIONAL ERROR LANDSCAPE VISUALISATION

To demonstrate the generality of our findings, we extend the error landscape visualisation to two additional, distinct datasets: DTD and SUN397 (Fig. 12). These visualisations complement the results shown in the main text and confirm that MIMA effectively aligns the task vector in the parameter space across different tasks.

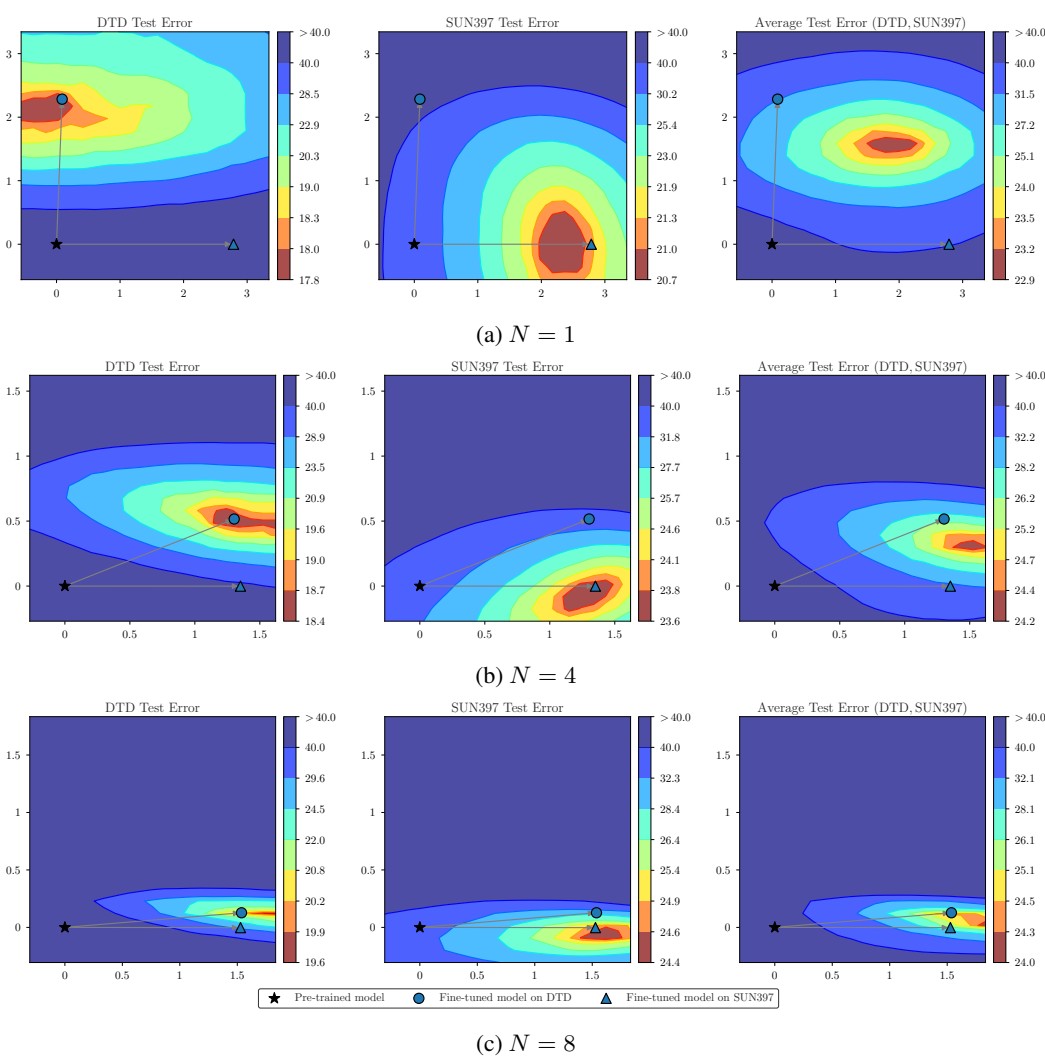

Figure 12: Error landscape visualisation for DTD (*left*), SUN397 (*middle*), and their average (*right*). Consistent with our findings in the main text, increasing the number of MIMA iterations ($N$) rotates the misaligned low-error valleys into a shared basin, providing the merged model with high performance.

# E THEORETICAL ANALYSIS: WHY MIMA IMPROVES ALIGNMENT

To gain theoretical insight into why iterative averaging and fine-tuning encourage task vector alignment, we analyse task vectors after the first and second iterations. For this theoretical analysis, we make two simplifying assumptions about the geometry of task-specific updates $\delta$:

1. (cross-task orthogonality) For any two distinct tasks $s \neq t$ and any iterations $i, j \in \{1, \ldots, N\}$, their respective updates are orthogonal:

$$\boldsymbol{\delta}_s^{(i)} \cdot \boldsymbol{\delta}_t^{(j)} = 0.$$

   This assumption reflects sufficiently high dimensionality of the parameter space, where updates for different tasks are unlikely to be correlated.

2. (within-task non-negativity) For each task $t$, the updates from any iterations $i, j \in \{1, \ldots, N\}$ exhibit non-negative alignment:

$$\boldsymbol{\delta}_t^{(i)} \cdot \boldsymbol{\delta}_t^{(j)} \geq 0.$$

   This assumption suggests that any fine-tuning step will not push the model outside the basin. In other words, the fine-tuning for a given task tends to move the parameters in a direction that is not entirely opposed to its other fine-tuning updates.

Under these assumptions, we can show that MIMA provably increases alignment.

**Proposition 1** (Improved Alignment via MIMA). *Under the assumptions of cross-task orthogonality and within-task non-negativity, for any two distinct tasks $i \neq j$, the pairwise cosine similarity between their task vectors after iteration 2 is greater than or equal to their cosine similarity after iteration 1:*

$$\cos\left(\boldsymbol{\tau}_i^{(2)}, \boldsymbol{\tau}_j^{(2)}\right) \geq \cos\left(\boldsymbol{\tau}_i^{(1)}, \boldsymbol{\tau}_j^{(1)}\right) = 0, \forall i \neq j.$$

*Proof.* By definition, task vectors after fine-tuning phase 1 are $\boldsymbol{\tau}_i^{(1)} = \boldsymbol{\delta}_i^{(1)}$ and $\boldsymbol{\tau}_j^{(1)} = \boldsymbol{\delta}_j^{(1)}$. Given the cross-task orthogonality assumption for $i \neq j$, we have:

$$\boldsymbol{\tau}_i^{(1)} \cdot \boldsymbol{\tau}_j^{(1)} = \boldsymbol{\delta}_i^{(1)} \cdot \boldsymbol{\delta}_j^{(1)} = 0.$$

Therefore, the cosine similarity between is:

$$\cos\left(\boldsymbol{\tau}_i^{(1)}, \boldsymbol{\tau}_j^{(1)}\right) = \frac{\boldsymbol{\tau}_i^{(1)} \cdot \boldsymbol{\tau}_j^{(1)}}{\|\boldsymbol{\tau}_i^{(1)}\|\|\boldsymbol{\tau}_j^{(1)}\|} = 0.$$

$$\boldsymbol{\tau}_i^{(2)} \cdot \boldsymbol{\tau}_j^{(2)} = \left(\frac{1}{T} \sum_{a=1}^{T} \boldsymbol{\delta}_a^{(1)} + \boldsymbol{\delta}_i^{(2)}\right) \cdot \left(\frac{1}{T} \sum_{b=1}^{T} \boldsymbol{\delta}_b^{(1)} + \boldsymbol{\delta}_j^{(2)}\right)$$

Applying the cross-task orthogonality assumption $\boldsymbol{\delta}_s^{(i)} \cdot \boldsymbol{\delta}_t^{(j)} = 0$, $\forall s \neq t$:

$$\boldsymbol{\tau}_i^{(2)} \cdot \boldsymbol{\tau}_j^{(2)} = \frac{1}{T^2} \sum_{a=1}^{T} \|\boldsymbol{\delta}_a^{(1)}\|^2 + \frac{1}{T}\left(\boldsymbol{\delta}_i^{(2)} \cdot \boldsymbol{\delta}_i^{(1)} + \boldsymbol{\delta}_j^{(2)} \cdot \boldsymbol{\delta}_j^{(1)}\right)$$

Since $\|\boldsymbol{\delta}_a^{(1)}\|^2 \geq 0$ for all $a$, and by the within-task non-negativity assumption, $\boldsymbol{\delta}_i^{(2)} \cdot \boldsymbol{\delta}_i^{(1)} \geq 0$ and $\boldsymbol{\delta}_j^{(2)} \cdot \boldsymbol{\delta}_j^{(1)} \geq 0$, it follows that:

$$\boldsymbol{\tau}_i^{(2)} \cdot \boldsymbol{\tau}_j^{(2)} \geq 0,$$

$$\cos\left(\boldsymbol{\tau}_i^{(2)}, \boldsymbol{\tau}_j^{(2)}\right) = \frac{\boldsymbol{\tau}_i^{(2)} \cdot \boldsymbol{\tau}_j^{(2)}}{\|\boldsymbol{\tau}_i^{(2)}\|\|\boldsymbol{\tau}_j^{(2)}\|} \geq 0 = \cos\left(\boldsymbol{\tau}_i^{(1)}, \boldsymbol{\tau}_j^{(1)}\right).$$

$\square$

Our theoretical analysis shows that this averaging and fine-tuning procedure is guaranteed to improve the cosine similarity between task vectors from the first iteration to the second iteration under assumptions of cross-task orthogonality and within-task non-negativity.

