# OpenReview forum: "MIMA: Iterative Model Averaging and Fine-Tuning for Multi-Task Learning"
_ICLR.cc/2026/Conference — Submitted to ICLR 2026_

### Official Review · Reviewer_fPJa · 2025-10-30

**Soundness:** 2
**Presentation:** 2
**Contribution:** 2
**Rating:** 2
**Confidence:** 4

**Summary:**

The paper proposes to interleave finetuning and merging to diminish degradation of performance on each task and keep the per task models aligned. Every finetuning step taken on each per task model is followed by a model averaging one, then the next finetuning step starts from the newly averaged one. Effectively it tries to keep per task models close to each other so merging them is easier.

**Strengths:**

- The idea is very straighforward and easy to implement
- Using the most current averaged model as starting point for the next finetuning step forces all fine-tuned models to be close to each other in the parameter landscape.
- The method seems to take care of the interference between tasks due to misalignment in parameters.
- It merges by averaging, so there is less hyperparameters to set compared to other methods.

**Weaknesses:**

- Very small experimental setting, the authors did not test other models beyond ViTs, for example since it was mentioned in the introduction, BERT, another transformer model. This makes the contribution less impactful.
- No details on how to set the number of steps for finetuning when tasks have different number of dataset sizes.

**Questions:**

- While this might work for easy to integrate tasks, like the 8 vision ones shown here that seem to play nice between each other, it would be interesting to see what happens when integrating one task comes at the cost of losing performance in others.
- In the task arithmetic code there are some suggestion for number of epochs (steps / batch size), which is even for the largest dataset lower than 20000 steps. This seems to be relevant for the method and is an hyper parameter. How it should be chosen? In particular when dealing with datasets of different sizes?

---

### Official Review · Reviewer_hvAW · 2025-10-31

**Soundness:** 3
**Presentation:** 3
**Contribution:** 3
**Rating:** 2
**Confidence:** 3

**Summary:**

This paper identifies that task vector orthogonality, arising from independent fine-tuning, is the primary cause of performance degradation in post-hoc model merging methods. To address this, the paper proposes MIMA (Multi-Task Iterated Model Averaging), an iterative framework that alternates between (1) averaging all task-specific models and (2) fine-tuning each from this shared starting point. The main contribution is the claim that this iterative process actively aligns the task vectors during the training phase, forcing their pairwise similarity to increase. This alignment enables the final merged model to achieve high multi-task performance using only simple, hyperparameter-free Weight Averaging.

**Strengths:**

- This paper distinguishes itself from prior model merging work that seeks to alleviate orthogonality post-hoc by directly intervening in the fine-tuning process to align task vectors, thereby addressing the problem at its root.

- The figures are intuitive and effectively convey the key ideas.

- The proposed method is simple and easy to reproduce.

**Weaknesses:**

- The method closely resembles a synchronized training paradigm similar to FedAvg. The difference is that each task keeps its own objective. This is useful but not fundamentally new. The paper does not provide theoretical analysis that explains why task vector alignment guarantees performance improvement. The contribution feels empirical rather than conceptual. This is useful but not fundamentally new.

- Frequent weight averaging increases memory and communication cost. This becomes a bottleneck for large foundation models.
No clear cost benefit analysis is given. It is unclear whether the method scales well in real deployment.

- A major weakness of this paper is restricted experimental scope (eight image classification tasks, which currently hinders the generalization of its claims. While the authors demonstrate the effectiveness of MIMA on vision tasks, model merging has also been actively explored in NLP. It thus remains unclear whether the central hypothesis—that independent fine-tuning produces nearly orthogonal task vectors and that MIMA can effectively align them—would hold for language models (like the cited BERT), or larger scale of vision tasks (14/20 vision tasks), or dense vision tasks (semantic segmentation, depth estimation, and surface normal estimation).

**Questions:**

- The authors claim that the goals of FedAvg and MIMA are different, but given that the paper shows the single merged model achieves nearly identical performance to the individual specialized models, why is it necessary to maintain the specialized models? This appears to be redundant and a waste of storage. Ultimately, this makes MIMA functionally indistinguishable from FedAvg's.

- Since the MIMA's aligned vectors differ from conventional task vectors used in prior merging approaches—by incorporating shared information—it would be interesting to see how recent model merging algorithms [1,2] behave when applied to these aligned representations.

[1] Task singular vectors: Reducing task interference in model merging, CVPR 2025
[2] Adamerging: Adaptive model merging for multi-task learning, ICLR 2024

---

### Official Review · Reviewer_Umwu · 2025-11-01

**Soundness:** 4
**Presentation:** 4
**Contribution:** 3
**Rating:** 4
**Confidence:** 3

**Summary:**

This paper introduces MIMA (Multi-Task Iterated Model Averaging), an iterative framework for multi-task learning that alternates between model averaging and task-specific fine-tuning. The central idea is that periodically synchronizing and averaging the weights of task-specific models helps align task vectors in parameter space, thereby reducing task interference and improving the performance of merged models. Experiments across eight vision datasets and various CLIP ViT architectures demonstrate that MIMA significantly narrows the single-task accuracy gap between fine-tuned and merged models, achieving near-perfect alignment as measured by cosine similarity between task vectors.

**Strengths:**

1. The paper is well-written and easy to follow.
2. The illustrations in Figure 1 (page 2) effectively show how independent fine-tuning leads to orthogonal task vectors and how MIMA mitigates this issue.
3. The experimental results clearly demonstrate improved model fusion performance as the number of fine-tuning rounds increases.

**Weaknesses:**

1. The proposed MIMA framework requires multiple full fine-tuning rounds on all tasks with supervised data. As a result, its computational and data requirements are comparable to traditional multi-task learning (MTL). The authors should compare their approach with standard MTL methods or alternative fine-tuning strategies that include model merging as a post-process.
2. I suggest including comparisons with mainstream MTL baselines, as well as reporting training costs, GPU hours, data requirements, and memory usage. MIMA may achieve performance comparable to strong MTL baselines, but possibly at a lower cost.
3. Although the paper claims a fixed total number of fine-tuning steps (S = 2000), the repeated averaging and synchronization phases may introduce additional overhead in distributed or large-scale settings. A computational cost analysis (e.g., GPU hours versus baseline fine-tuning) would help substantiate claims of efficiency.
4. The proof in Section E only covers two iterations under idealized assumptions. Extending the analysis to general N > 2 or relaxing some assumptions would enhance the theoretical credibility of the work.

**Questions:**

Please see the weaknesses.

---

### Official Review · Reviewer_gRTD · 2025-11-01

**Soundness:** 2
**Presentation:** 3
**Contribution:** 2
**Rating:** 2
**Confidence:** 4

**Summary:**

The manuscript proposes MIMA, an iterative merging framework that alternates between task-specific fine-tuning and model averaging.
The goal is to align vectors based on cosine similarity to improve performance when merging several STL models for MTL. MIMA aims to improve alignment via. repeated averaging and refine-tuning, leading to higher cosine similarity and better merged model performance on several vision benchmarks.

**Strengths:**

The paper has a clear set-up.

The authors propose a simple and general framework, involving an iterative averaging procedure which is straightforward and compatible with existing merging techniques (Task Arithmetic, Ties-Merging).

The results include cosine similarity measurements and correlations between task alignment and performance.

**Weaknesses:**

The conceptual novelty is there but somewhat limited. The core MIMA algorithm closely mirrors Federated Averaging (FedAvg) and other iterative averaging schemes (e.g., Model Soup, WiSE-FT). The difference lies in optimizing for alignment rather than global convergence but is mostly interpretive. However, the use of cosine similarity as a proxy for task alignment is intuitive but not theoretically grounded. The literature on alignment metrics and generalization is not discussed (see for example arXiv:1809.10374).

The paper lacks a formal theoretical justification or convergence analysis explaining why iterative averaging should promote beneficial alignment.

Task Arithmetic (TA) does not use fine-tuning or data access during merging, whereas MIMA repeatedly fine-tunes with data. This makes the comparison somewhat imbalanced, as MIMA consumes additional compute and supervision. Any claim that MIMA 'improves upon TA' must be framed accordingly. Furthermore, the joint MTL literature, which has come up with many ways to improve task alignment (even if during joint training) is ignored.

No experiments compare MIMA to joint multi-task training (i.e., training a single model on all tasks together), which seems like a logical baseline.

`The literature review is limited and outdated. The discussion of federated learning cites only McMahan et al. (2017), an 8-year-old reference.

The paper overlooks modern adaptive weighting and gradient surgery approaches (the strand of work including e.g., Yu, T., Kumar, S., Gupta, A., Levine, S., Hausman, K., & Finn, C. (2020). Gradient surgery for multi-task learning. Advances in neural information processing systems, 33, 5824-5836. and follow-up work).

No heterogeneous tasks or modalities are evaluated. All experiments use similar vision benchmarks with CLIP-based encoders.

**Questions:**

The paper claims that MIMA reduces the need for validation sets because merging methods converge to similar performance. But MIMA introduces new degrees of freedom (number of iterations, averaging strategy, etc.) that themselves require validation.

It is unclear why only encoder layers were fine-tuned while decoder layers remained fixed.`

Do you have experimental results for more diverse sets of conditions (task diversity).

Could you provide a deeper discussion on alignment metrics and generalization?

Is there a formal theoretical motivation behind your approach?

What is the main practical scenario you would use this method for? As it requires access to data.

Would it make sense to first merge and then MTL fine-tune with (one of) many of the SOTA dynamic weighting algorithms from MTL literature?

What would the impact of heterogeneous or higher dimensional tasks be?

**Details Of Ethics Concerns:**

No concerns

---

### Meta-Review · Area_Chair_zWmR · 2026-01-07

**Summary:**

The paper introduces MIMA, an iterative framework for multi-task learning that alternates between task-specific fine-tuning and model averaging to align task vectors in parameter space.  Experiments focus on eight vision benchmark tasks using CLIP-based ViT architectures, showing improved alignment and merged performance while retaining strong single-task accuracy.
Reviewers generally appreciate the clarity of the presentation, intuitive figures, and the straightforward nature of the method. However, the novelty of this manuscript is limited (the similarity to FedAvg and other federated methods), lack of theoretical grounding or convergence analysis, and absence of comparisons to standard multi-task learning (MTL) baselines or joint training. Reviewers also note incomplete literature coverage, particularly on modern MTL techniques like gradient surgery or adaptive weighting.

**Reviewer Concerns:**

No explicit author rebuttal or response was posted in the forum.

**Reviewer Scores:**

Given the lack of an explicit rebuttal or discussion phase evident in the forum, I believe the scores would remain largely unchanged.

---

### Decision · Program_Chairs · 2026-01-26

Reject